# A Helical Microrobot with an Optimized Propeller-Shape for Propulsion in Viscoelastic Biological Media

**Dandan Li** [1,2], **Moonkwang Jeong** [2], **Eran Oren** [3], **Tingting Yu** [1,2] **and Tian Qiu** [1,2,*] 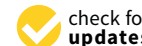

[1]  Cyber Valley Research Group, Institute of Physical Chemistry, University of Stuttgart, Pfaffenwaldring 55, 70569 Stuttgart, Germany; lidan@is.mpg.de (D.L.); yu@is.mpg.de (T.Y.)

[2]  Micro, Nano, and Molecular Systems Group, Max Planck Institute for Intelligent Systems, Heisenbergstrasse 3, 70569 Stuttgart, Germany; jeong@is.mpg.de

[3]  Bionaut Labs Ltd., Los Angeles, CA 90034, USA; eran.oren@bionautlabs.com

*  Correspondence: tian.qiu@ipc.uni-stuttgart.de

**Abstract:** One major challenge for microrobots is to penetrate and effectively move through viscoelastic biological tissues. Most existing microrobots can only propel in viscous liquids. Recent advances demonstrate that sub-micron robots can actively penetrate nanoporous biological tissue, such as the vitreous of the eye. However, it is still difficult to propel a micron-sized device through dense biological tissue. Here, we report that a special twisted helical shape together with a high aspect ratio in cross-section permit a microrobot with a diameter of hundreds-of-micrometers to move through mouse liver tissue. The helical microrobot is driven by a rotating magnetic field and localized by ultrasound imaging inside the tissue. The twisted ribbon is made of molybdenum and a sharp tip is chemically etched to generate a higher pressure at the edge of the propeller to break the biopolymeric network of the dense tissue.

**Keywords:** microrobotics; micro-propulsion; viscoelasticity; biological tissue; microfabrication

---

## 1. Introduction

Micro-/nano-robotics hold great potential in biomedical applications [1]. They are minimally invasive, leaving minimal, or even no, surgical footprint. They can be wirelessly driven and controlled, which should enable a number of medical applications, including drug delivery, and in vivo sensing and stimulation, especially if they can also be navigated into tissues. They may also enable new surgical procedures that treat non-surgical diseases, such as infection or immune responses at the cellular level. To accomplish these biomedical tasks, the premise is to develop suitable propulsion strategies for microrobots in complex biological media.

Most biological media are non-Newtonian [2]. The fluids exhibit complicated viscoelastic properties, and their response depends on the shape and the size of the device and its mode of actuation, such as the frequency of operation and the shear rate. It is therefore important to develop suitable mechanisms for the propulsion in a biological medium. Propulsion of micro-/nano-robots has so far mainly been demonstrated in Newtonian fluids, such as water; or in low-viscosity bodily fluids, such as in the gastrointestinal tract [3,4] and in the abdominal cavity [5]. Micro-propulsion in biological viscoelastic fluids remains relatively unexplored. An exception is a clamshell-like "microscallop" that modulates the local fluid viscosity upon varying the motion speed, so that it can swim by simple reciprocal motion in shear-thinning biological fluids [6]. Recently, the first nanorobots that can penetrate real biological tissue have been reported. The helical nanorobots have a diameter of ~500 nm, which is similar to the mesh size of the biopolymer network in the vitreous of the eye, thus they can propel

through the vitreous and precisely target the macular region of the retina [7]. Compared to nanorobots, micro-sized robots have a much larger volume, and offer a much higher capacity to carry onboard payloads for biomedical applications. For example, they can carry a higher dosage of drug for drug delivery or take useful electronic integrated circuits onboard. However, larger microrobots can no longer simply 'slip' through the macromolecular tissue network. Here, we experimentally show that a particular design and shape of a microrobot of hundreds of micrometers in diameter can be actively propelled through dense liver tissue.

Helical propellers are commonly adopted to propel microrobots at a small scale, as they transform a rotation around their helical axis into a linear translation [8]. Early works have shown that a helical screw-shape propeller can drill through elastic tissue, such as bovine muscle [9], a pork fillet [10] or a blood clot [11]. However, these devices are large with a typical diameter of several millimeters and a length of several centimeters. For much more minimally-invasive procedures, it is beneficial to reduce the size of the device, especially the diameter, to the regime of sub-millimeter. This poses extra challenges for the wireless actuation of the microrobots. As the actuation is realized by a rotating magnetic field [12], the magnetic torque applied on the robot is proportional to its magnetic moment, which scales with the volume ($L^3$) of the magnetic part of the robot; but the fluidic drag scales with the surface area ($L^2$) of the robot, where $L$ is the characteristic length of the device. As the size of the device is reduced, it becomes more and more difficult to provide enough magnetic torque to overcome the drag and achieve propulsion, especially in dense tissues. Therefore, careful design of the shape of the micropropeller is required to enable the propulsion.

In this paper, we report that a special "twisted" helical shape together with a high aspect ratio in cross-section permits a microrobot that is hundreds of micrometers in diameter to penetrate mouse liver (Figure 1), when other propeller designs fail. The microrobot can be localized by ultrasound imaging. A sharp tip is prepared by chemical etching, which results in a higher pressure at the tip and leads to a higher propulsion speed.

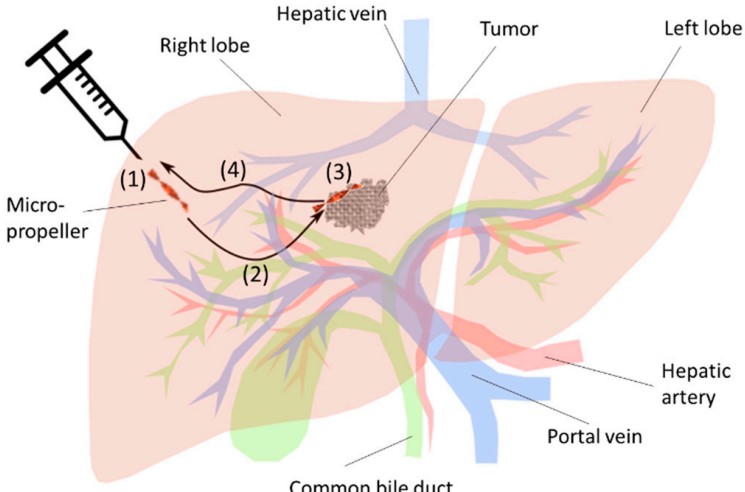

**Figure 1.** Schematic of the helical microrobot used as a minimally-invasive delivery vehicle in the biological tissue. The propeller is not drawn to scale. The delivery steps include: (1) Injection of the robot into the organ; (2) Active propulsion and navigation to avoid important anatomical structures and reach the vicinity of a lesion, e.g., a tumor; (3) Drug release at the targeted location; (4) Retrieval of the robot.

## 2. Materials and Methods

### 2.1. Fabrication of the Micropropellers

The twist-shape micropropellers were fabricated by the following process, as illustrated in Figure 2. Molybdenum (Mo) foil (50 μm thick) was cut by laser (MPS Flexible with the laser StarFemto FX,

ROFIN-BAASEL Lasertech GmbH, Germany) into rectangular plates with a length of 50 mm and a width of 0.5 mm. The plate was fixed to a customized rotational stage, where the bottom clamp was stationary and the top clamp was rotated. During rotation the clamp distance was reduced, until the desired pitch (~4 mm) was reached. Each individual propeller was mechanically cut into a length of ~2 mm, which consists of about half-pitch. A cylindrical magnet (0.2 mm in diameter and 2 mm in length, magnetized in the diametric direction, GMB Deutsche Magnetwerke GmbH, Germany) was attached to the end of the propeller using cyanoacrylate glue (UHU GmbH, Germany).

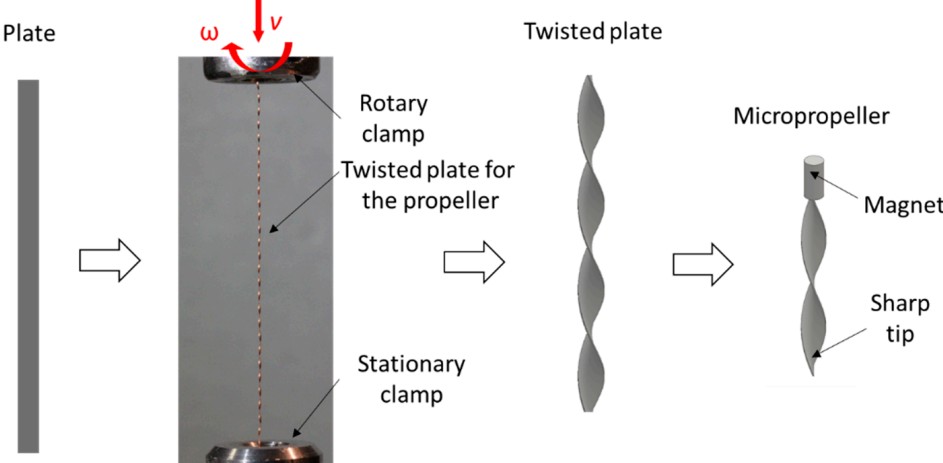

**Figure 2.** Fabrication process of the micropropeller. A metallic plate is cut into the designed width by laser, mechanically twisted to form a helix with the designed pitch, and cut into multiple propellers. A sharp tip is chemically etched at one end and a magnet is attached at the other end to assemble the micropropeller.

To fabricate the screw-shape micropropellers, the Mo foil (50 μm thick) was cut by laser into rings (outer diameter 0.7 mm, inner diameter 0.3 mm) with a thin gap for further stretching. The two ends of the Mo ring at the gap were clamped and mechanically stretched in the opposite direction along the central axis with a distance of ~0.5 mm, which represents the helical pitch of ~0.5 mm. Two stretched rings was glued onto a cylindrical magnet (0.2 mm in diameter and 2 mm in length, magnetized in the diametric direction, GMB Deutsche Magnetwerke GmbH) to fabricate a continuous helical shape using cyanoacrylate glue.

*2.2. Etching of the Sharp Tips*

As shown in Figure 3, the tip of the micropropeller was chemically etched, similar to the etching procedure used to etch the probe for scanning tunneling microscopy [13]. The twisted Mo plate worked as the anode and was dipped into the electrolyte of 2 M NaOH, with a depth of 1–2 mm. The cathode, a silver wire (0.5 mm diameter) bent into a ring of 4 cm diameter, was immersed in the electrolyte around the twisted plate anode. The etching voltage applied between the Mo anode and the silver cathode was controlled by a DC power supply (PeakTech GmbH, Germany). When the immersed metal part detached from the anode, the etching was stopped immediately and the tip was rinsed with deionized water. The etched tip was imaged by a scanning electron microscope (SEM, Ultra 55, Carl Zeiss AG, Germany) and is shown in Figure 3.

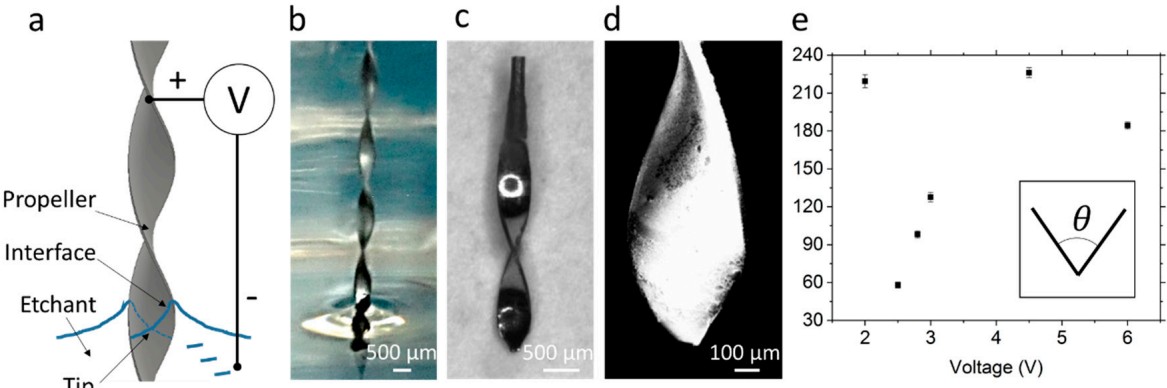

**Figure 3.** Chemical etching of the tip of the micropropeller. (**a**) Schematic of the chemical etching process; (**b**) a picture of the twisted plate during etching; (**c**) microscopic image of the micropropeller; (**d**) SEM image of the sharp tip; (**e**) the etched tip angle versus etching voltage. Error bars represent standard deviations.

### 2.3. Magnetic Actuation Set-up

To provide enough magnetic torque for the microrobot actuation in soft elastic tissue, a rotating magnetic field with a high field strength and a large working volume is required. We built a magnetic actuation set-up based on the assembly of four rotating permanent magnets, and demonstrated that it can generate strong oscillating magnetic fields for linear actuation [14] or a rotating magnetic field that is homogenous in amplitude for rotational actuation [15]. Briefly, four sets of cubic magnets (assembled from eight Neodymium N45 magnets of $30 \times 30 \times 15$ mm$^3$ each, Supermagnete, Webcraft GmbH, Germany) were driven by an electric servomotor (BMH0702T11A1A with an embedded encoder and a gear box GBX040003K, Schneider Electric GmbH, Germany) and rotated in the same direction along the axis perpendicular to the magnetization, synchronized by a timing belt-pulley mechanism. The rotational speed of the motor was controlled by computer software (SoMove, Schneider Electric GmbH) and the controller (LXM32AD18M2, Schneider Electric GmbH). A 3-axis magnetometer (3MH3A-0.1%-200 mT, Hall Probe C with F3A transducer, Senis AG, Switzerland) was used to measure the magnetic field in the working volume of the set-up. A magnetic field of ~100 mT with a homogeneity of ±10% was achieved in a working volume of $20 \times 20 \times 20$ mm [15].

### 2.4. Propulsion Tests

Matrigel (Geltrex™, Gibco®, Thermo Fisher Scientific, Germany) was prepared as a model viscoelastic medium to test the microrobots. The frozen Matrigel was thawed at 4 °C, filled in a plastic container, and gelled at 37 °C for 1 h. The microrobot of three different geometries, i.e., screw-shape (outer diameter 0.7 mm, length 2 mm, pitch 0.5 mm), twist-shape with a flat tip (outer diameter 0.5 mm, total length 4 mm, pitch 4 mm), twist-shape with a sharp tip (outer diameter 0.5 mm, total length 5 mm, pitch 4 mm), were used for the propulsion test. It was pushed into the gel severally with a pair of tweezers and the container was placed in the center of the working volume of the magnetic actuation system. To compare three different geometries of the propellers, a fixed rotational frequency of 2 Hz was applied. To test the frequency dependency of the twisted propeller, the frequency was set at 0.5, 1.0, 1.5, 1.75 and 2 Hz, respectively. The test at each frequency was repeated in new samples at least three times. Videos were taken by a camera (600D with an EF 100 mm f/2.8 L macro lens, Canon, Japan) at a frame rate of 30 fps. A sequence of frames was extracted for each video and analyzed in ImageJ (Fiji 1.52p, NIH, USA) to calculate the propulsion speed.

The propulsion experiments were conducted in BALB/c mice obtained after euthanization and conducted within 1 h after euthanization. Experimental procedures were performed following the ethical approval (No. 35-9185.81/G-18/84 issued by the Regional Council Freiburg, Germany). An incision was made on the abdominal skin, and the liver was exposed. A pair of tweezers was used

to insert the propeller into a lobe of the liver. The propeller was actuated by a rotating magnetic field at the frequency of 3 Hz. Ultrasound gel was applied on top of the liver and a linear array ultrasound transducer (L11, GE Healthcare, Chicago, Illinois, USA) was used for imaging. Videos were recorded by an ultrasound imaging machine (12 MHz, LOGIQ P6, GE Healthcare).

*2.5. Rheology Measurement*

The rheology of the Matrigel was measured on a rotational rheometer (Kinexus Pro, Malvern Instruments GmbH, Germany) with a plate–plate geometry of 20 mm in diameter and a gap of 0.7 mm. The Matrigel solution was added in between the plates and cured at 37 °C for 30 min. A shear strain sweep in the range of 0.01–100% was carried out in an oscillation test at the frequency of 1 Hz. A shear stress ramp from 1 Pa to 100 Pa was conducted. All measurements were carried out at 37 °C.

## 3. Results

*3.1. Fabrication of the Micropropellers*

Microrobots of three different shapes, i.e., screw-shape (outer diameter 0.7 mm, length 2 mm, pitch 0.5 mm), twist-shape with a flat tip (outer diameter 0.5 mm, total length 4 mm, pitch 4 mm), twist-shape with a sharp tip (outer diameter 0.5 mm, total length 5 mm, pitch 4 mm), were fabricated. The cylindrical permanent magnet is included as a shaft in the screw-shape, and it is attached to the end of the twist-shape. The sharp tip of the metallic propeller was fabricated by electrochemical etching. Due to the concave air-liquid interface (Figure 3a,b), a convective flow was formed and the metal was etched faster at a position underneath the surface, so that a sharp tip was formed [13]. To fabricate a sharp tip at the end of the complicated geometry of a twisted plate, we optimized the DC voltage applied for etching. When the voltage is too low or too high, the etching results in an almost flat end. At a voltage of 2.5 V, the tip angle $\theta$ is the smallest at ~60° for a twisted Mo plate of 0.5 mm in width (Figure 3c–e). The resulting tip size is ~50 µm measured by SEM. The chemical etching not only results in a sharp tip, but also thins the edges near the tip, which also benefits the cutting at the tip front and facilitates propulsion in tissue.

*3.2. Rheological Results and Modelling of the Matrigel*

The Matrigel was characterized in an oscillation test at the frequency of 1 Hz (Figure 4a). In the scanned strain range of 0.1–100%, the elastic modulus G′ is more than an order of magnitude higher than the viscous modulus G″, which suggests that Matrigel has a gel-like elastic-dominant behavior. The linear viscoelastic (LVE) is up to ~20% strain. The peaks of G′ and G″ at high strain is typical for cross-linked polymers. The viscous modulus G″ represents the deformation energy, which is dissipated due to inner friction processes. Increasing G″-values indicate an increasing portion of deformation energy before the final breakdown of the internal superstructure occurs [16]. Leaving the LVE plateau at a strain of 20–100% suggests that irreversible deformation of the gel occurs, i.e., the gel network starts to break down.

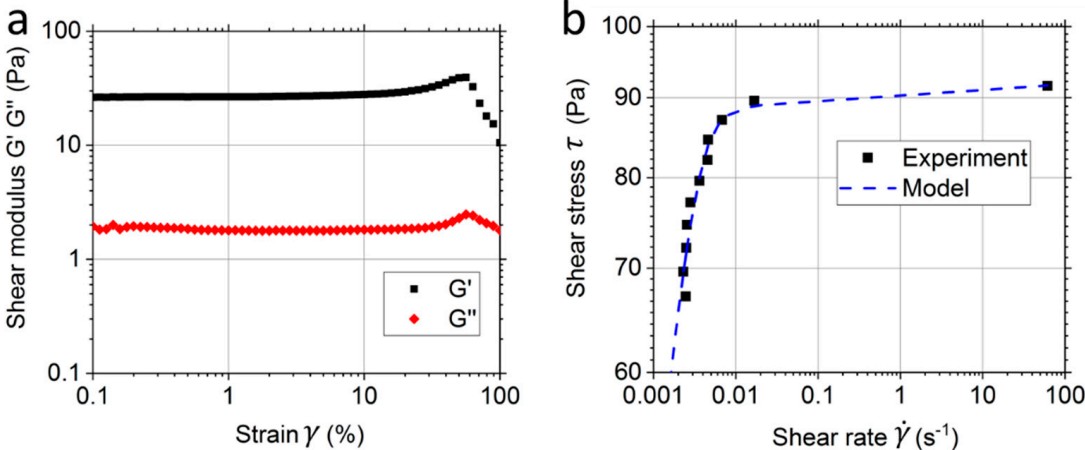

**Figure 4.** Rheological measurement and modelling of Matrigel. (**a**) Strain sweep in an oscillation test at the frequency of 1 Hz; (**b**) shear rate is measured during a shear stress ramp in a rotational test. A Herschel–Bulkley–Papanastasiou model was used to fit the data.

The Matrigel was also measured in a rotational test with a shear stress ramp (Figure 4b). The data was fitted to a viscoplastic Herschel–Bulkley–Papanastasiou model [17], in which the relationship between the shear stress $\tau$ and the shear rate $\dot{\gamma}$ follows the equation:

$$\tau = \tau_y\left[1 - \exp\left(-m\dot{\gamma}\right)\right] + K\dot{\gamma}^n.$$

where the fitting parameters are yield stress $\tau_y = 76$ Pa, and the coefficients $m = 603$, $n = 0.02$, and $K = 14$. The yield stress is comparable to the elastic modulus of ~100 Pa reported in the literature [18]. The yield stress point of the Matrigel corresponds to a shear strain of 93%, which is consistent with the result from the oscillation test. Both rheological tests reveal the break-down limits of the gel, and suggest that in order to effectively propel in the viscoelastic gel, the propeller should allow higher stress at the tip over the yield stress, while maintaining lower stress along the propeller body below the yield stress.

### 3.3. Propulsion in the Matrigel

All three micropropeller designs (Figure 5a) are able to synchronously rotate with the rotating magnetic field in the Matrigel, suggesting that the magnetic torque is high enough to overcome the drag. When they are rotating at the same frequency of 2 Hz, the propulsion speeds are compared in Figure 5b,c (see also Supplementary Videos S1–S3). The traditional screw-shape has the lowest speed of 6 μm/s. The twist-shape propellers are at least five times faster, i.e., 35 μm/s and 51 μm/s for the ones without and with sharp tips, respectively. It shows that the twist-shape is superior to the screw-shape design for the propulsion in the Matrigel, a viscoelastic solid. The aspect ratio of the blades' cross-section, defined as the ratio of the largest radius $L_{max}$ and the smallest radius $L_{min}$ of the propeller, differs for the different designs, as can be seen in the cross-sectional view shown in Figure 5a. Tracer particles in the fluid show that the fluid near the propeller behaves very differently in the viscoelastic solid compared to the viscous fluid. In a viscous fluid, the fluid flows around the propeller together with the propeller. However, in the viscoelastic solid, the medium around the propeller separates from the rest of the tissue and forms a "plug" around the propeller. The plug adheres firmly to the propeller and rotates with the propeller at the same speed. The rest of the medium then experiences very little strain, causing the propeller to "freely" spin without generating any forward motion. For a traditional screw-shaped propeller, the aspect ratio is relatively small. When the viscoelastic gel breaks from the medium due to the rotation of the propeller, a large amount of the gel is observed to adhere to the propeller and rotates together with it, resulting in an almost cylindrical structure, which does not induce a large deformation on the rest of the matrix around the rotating structure. Therefore, it limits the propulsion force in the axial direction. When the aspect ratio of the cross-section in the

twist-shape design is increased to ~10:1, a "plug" is also formed, but the propeller with the plug is still helical and can nevertheless provide a propulsion force. The frequency dependency of the optimized design, i.e., the twist-shape propeller with the sharp tip, was also studied and is shown in Figure 5d. The propulsion velocity increases linearly with the frequency from 0.5 to 2 Hz.

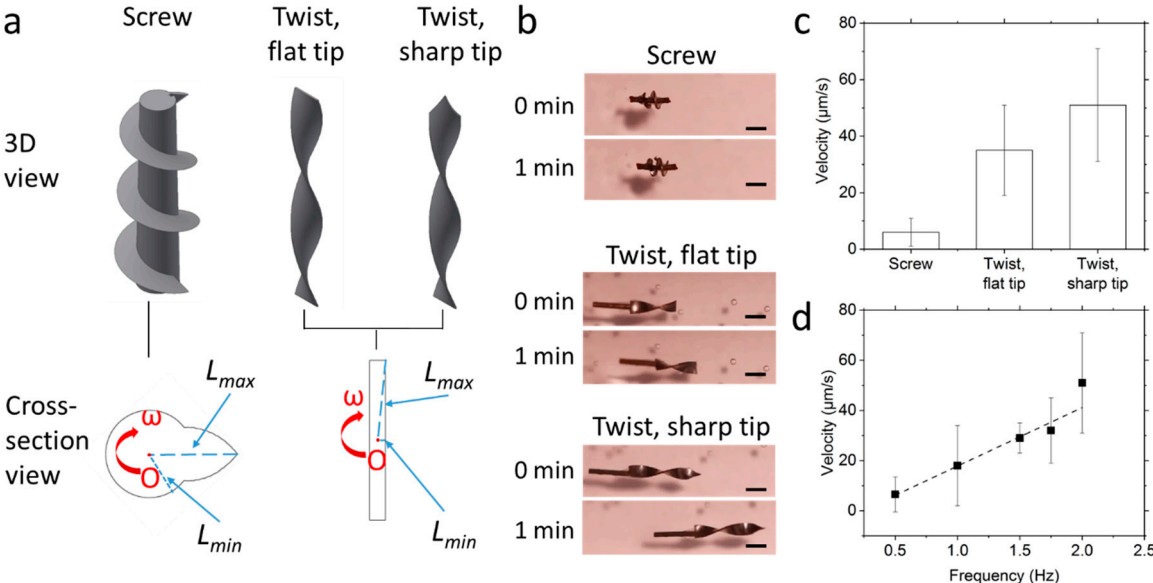

**Figure 5.** Propulsion of the micropropeller in the Matrigel. (**a**) Comparison of the traditional screw-shape propeller to the current twist-shape propellers with or without a sharp tip; (**b**) snapshots of the videos showing that the twist-shape propeller with the sharp tip achieves the fastest propulsion speed among the three kinds. Scale bars are 1 mm; (**c**) velocity of the three kinds of micropropellers rotated at 2 Hz; (**d**) the frequency response of the twist-shape propeller with the sharp tip. Error bars represent standard deviations.

### 3.4. Propulsion in the Mouse Liver

Propulsion tests of the microrobot were also carried out in a mouse liver as a real elastic tissue (Figure 6a). Ultrasound imaging shows the propulsion clearly (Figure 6b and Supplementary Video S4), and the average propulsion speed is 10 ± 2 μm/s. It is lower than the speed in the Matrigel, as the liver has a higher elastic modulus of ~600 Pa [19], and is more difficult to penetrate. When the rotational direction of the magnetic field is reversed, the propeller can reverse the propulsion direction and exit the tissue following the same trajectory. The reverse propulsion speed is higher than the forward speed, as a channel is already formed in the elastic tissue. The microrobot can also penetrate the entire lobe for a distance of ~8 mm and exit from the other side (Figure 6c,d). Some bleeding was seen at the injection spot, indicating that the tissue was still fresh. At the exit point, the tip of the microrobot penetrates the liver capsule, and a pair of tweezers were used to retrieve it from the tissue. The propulsion results were reproducible with four robots in different lobes of two mice.

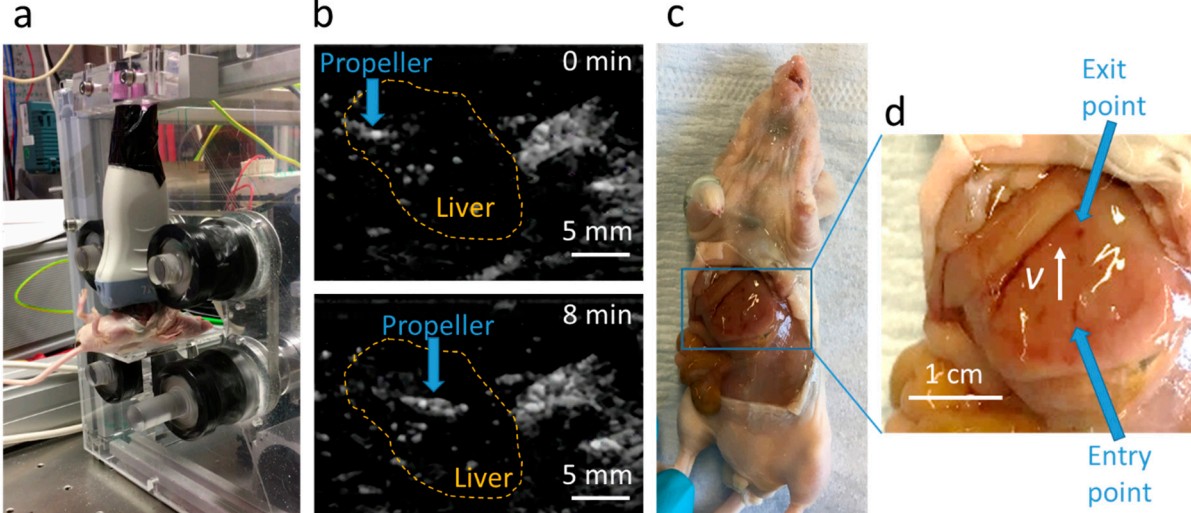

**Figure 6.** Propulsion of the microrobot in a mouse liver. (**a**) Experimental set-up for the magnetic actuation and ultrasound imaging. (**b**) Ultrasound images show clear propulsion in the mouse liver. (**c,d**) Dissection of the mouse showing the entry and exit points of the propeller in the main lobe of the liver. The white arrow indicates the propulsion direction.

## 4. Discussion

Helical structures are commonly used as propulsion units for micro-/nano-robotics. They are fabricated by various techniques. For example, for very small ones, photolithography and lift-off process [20], glancing angle deposition [21] and two photon lithography [22] have been adopted. For millimeter-diameter propellers, machining [9] and 3D printing [23] have been applied. To the best of our knowledge, there has been no report in the literature of using the twist method to make a sub-millimeter helical structure. Compared to the other previous methods, the new method reported here can fabricate metallic structures with hundreds of micron resolution. The sub-millimeter diameter is designed to be minimally invasive to the organ, while it is able to carry useful onboard payloads, such as a drug capsule or electronics. In the liver penetration experiments, tissue damage is confined only along the penetration trajectory of the micropropeller, and histological examinations will be conducted in future to evaluate the tissue damage around the microrobot.

The metal Mo has very high Young's modulus and hardness, and it is not brittle, so it is suitable to be mechanically twisted into a helical structure. The helical structure also helps to enhance the rigidity in the axial direction, making the robot more stable during the insertion and propulsion processes in the tissue. The Mo provides strong acoustic contrast and the propeller is clearly visible, even deeper in the tissue. The tip is very sharp and etched down to tens of micrometers. Experiments also show that the microrobots with sharp tips and edges propel faster than the ones without.

Matrigel is used as a model for viscoelastic biological tissue, as it is a commonly used gel medium for three dimensional cell culture [24]. Matrigel mostly consists of laminin and collagen, which provides a good mimic of in vivo extra cellular matrix (ECM) [25]. Rheological tests show that the gel is mainly elastic, but the gel network breaks down at low shear strain and stress. The traditional screw-shape propeller does not propel efficiently in the viscoelastic media. The reason is that a plug of the gel is formed around the propeller, which prevents propulsion. In the current twist-shape design, the high aspect ratio in the cross-section keeps the helical shape of the propeller, even if material adheres and forms a plug around the propeller. The new shape shows much faster propulsion speeds. Experiments show that the propulsion direction can be reversed in the tissue by switching the rotational direction of the magnetic field. It is also possible to steer the microrobot by controlling the direction of the rotational axis of the external magnetic field, as the robot's magnetic moment tends to align with the external field. However, it is more challenging in a viscoelastic solid than in a viscous fluid, as the resistance in

the medium against the steering is much higher. Future work will investigate the maneuverability of the microrobot, such as the minimal steering radius and the steering accuracy.

We report here the initial success of a controlled micro-sized robot propelling in viscoelastic solids. The design not only works in the Matrigel as a model fluid, but also in a mouse liver. Our design will serve as the basis for the further optimization of the microrobots' shape and material for the penetration of biological viscoelastic tissues.

## 5. Patent

A patent related to the micropropeller in viscoelastic biological tissue (application no. EP17166356.0) is pending.

**Supplementary Materials:** The following are available online at http://www.mdpi.com/2218-6581/8/4/87/s1, Video S1: The traditional screw-shape microrobot moves very slowly in the Matrigel. The video is in real time. Video S2: The twist-shape microrobot with the flat tip moves in the Matrigel. The video is in real time. Video S3: The twist-shape microrobot with the sharp tip moves fast in the Matrigel. The video is in real time. Video S4: Ultrasound imaging of the microrobot propelling in a mouse liver. The video is sped up 100 times.

**Author Contributions:** Supervision, T.Q.; Propeller fabrication, D.L.; Magnetic actuation system, M.J.; SEM, Y.T.; Propulsion tests, D.L., E.O. and T.Q.; Data Analysis, D.L.; Writing—Review & Editing, D.L., M.J. and T.Q.

**Funding:** This research was partially funded by the Vector Foundation and the Max Planck Society.

**Acknowledgments:** The authors are grateful to Peer Fischer for many helpful discussions and support. The authors thank Thomas Meisner, Arnold Weible and the mechanical workshop in the Max Planck Institute for the help in the fabrication of the propeller.

**Conflicts of Interest:** One of the authors (T.Q.) is an inventor of a pending patent on the micropropeller. The authors declare no other conflicts of interest. The funders had no role in the design of the study; in the collection, analyses, or interpretation of data; in the writing of the manuscript, and in the decision to publish the results.

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
