# Peer review of "A Helical Microrobot with an Optimized Propeller-Shape for Propulsion in Viscoelastic Biological Media"

_robotics, doi:10.3390/robotics8040087_

Round 1
Reviewer 1 Report
This paper presents a microrobot with a diameter of hundreds of micrometers to move through mouse liver tissue by optimizing the propeller twist and cross-sectional aspect ratio. The paper is well organized and presented. The helical microrobot provides a promising option for the propulsion in viscoelastic biological medium. Overall, I suggest minor revisions from authors before publication. Here are some comments to improve the paper.
1. One thing confused is that the type of the paper should be a “research paper” rather than a “Review”, which is marked on the top of the title. This should be corrected.
2. The fabrication method of the helical microrobot is very interesting, simple and effective. Is there any similar approach presented in the literature? Could authors discuss more about this matter?
3. In the experiments, how to determine the rotation frequency of the micropropellers? Such as the 1 Hz or 2 Hz?
4. In the liver penetration experiment, was the evaluation about tissue damages conducted?
Reviewer 2 Report
This article reports a helical microrobot which can propel in viscoelastic biological Media. This work improves upon the current state-of-the-art in magnetic microrobots which can propel through biological tissues.
The authors have claimed that they optimized the propeller twist and cross-sectional aspect ratio of the helical microrobot which allowed actuating the microrobot through mouse liver tissue. Although the results are very promising, the analysis to support the claim is buried in the manuscript. Authors are suggested to explicitly discuss the optimization in light of the previous results upon which this work is optimized. Alternatively, the claim may re-worded in the abstract to reflect the actual results obtained in this work. In line # 42, the authors commented that larger microrobot has difficulties slip through macromolecule tissues. It is suggested to qualify the comments mentioning, why larger microrobot is even required when smaller can navigate through the environment. In sec. 2.1, microrobot fabrication is confusing. In the first paragraph, twisting MO foil using a pair of clamps is mentioned fabricating a micropropeller 0.5 mm diameter, ~2 mm length and ~4 mm pitch. Does it mean the microrobot length contains half-pitch? The second paragraph mentions different fabrication technique of micropropeller, which is more confusing. Please clarify.
Please cite the figure no. in line 88.
Please mention the microrobot size in sec. 2.4. Also, mention the viscosity of the Matrigel.
Please mention the dimensions of each microrobot in sec. 3.1. It is mention is a couple of places that the MO plate is of 0.5 mm in width, does it mean the thickness of MO sheet is 0.5 mm? Or, the micropropeller width is 0.5 mm. Please, clarify and edit in the manuscript. In sec. 3.3, is the microrobot actuating in gelled Matrigel? Also, by saying viscous fluid and viscoelastic solid what medium are referred here? Authors are requested to explain clearly the “plug” behavior of the different shape microrobots actuation in sec. 3.3. Authors are suggested to include a brief discussion about the steering of the microrobots including steering accuracy, and resolution. Please add scale bar in S1- S3. Please incorporate the revisions in the manuscript and highlight the edits.
Round 2
Reviewer 2 Report
Thank you for addressing my comments and concern.